# Penetration of Enrofloxacin in Aqueous Humour of Avian Eyes

**DOI:** 10.3390/vetsci10010005

**Published:** 2022-12-23

**Authors:** Katrin Fuchs, Monika Rinder, Richard Dietrich, Leena Banspach, Hermann Ammer, Rüdiger Korbel

**Affiliations:** 1Center for Clinical Veterinary Medicine, Clinic for Birds, Small Mammals, Reptiles and Ornamental Fish, Veterinary Faculty, University of Munich, 85764 Oberschleissheim, Germany; 2Chair for Hygiene and Technology of Milk, Department of Veterinary Sciences, Veterinary Faculty, University of Munich, 85764 Oberschleissheim, Germany; 3Department of Veterinary Drug Residues, Bavarian Health and Food Safety Authority (LGL), 91058 Erlangen, Germany; 4Institute of Pharmacology, Toxicology and Pharmacy, Department of Veterinary Sciences, Veterinary Faculty, University of Munich, 80539 Munich, Germany

**Keywords:** enrofloxacin, pharmacokinetics, birds, chicken, intraocular bacterial infections

## Abstract

**Simple Summary:**

Scientific information on the suitability of antibiotics for the treatment of intraocular bacterial infections in birds is lacking so far. Because the anatomical features of avian eyes differ from those in mammals, data obtained in mammals cannot be simply transferred to birds. Enrofloxacin, which is known to be a useful drug for the treatment of bacterial eye infections in mammals, was tested in chickens serving as a model system for birds. Our pharmacokinetic investigations revealed that single intramuscular administration of enrofloxacin at a dosage of 25 mg/kg body weight produced sustained and therapeutically active levels of enrofloxacin in the aqueous humour of chicken eyes.

**Abstract:**

Enrofloxacin has been shown to be appropriate to treat bacterial eye infections in mammals. However, the anatomy and physiology of the avian eye substantially differ from those in mammals, and pharmacokinetic data substantiating the clinical efficacy of enrofloxacin in birds are still lacking. In total, 40 chickens (*Gallus gallus*, Lohman Selected Leghorn) received single intramuscular administration of enrofloxacin at a dosage of 25 mg/kg body weight (BW). Serial blood and aqueous humour samples were taken at 12 different time points after administration (0–60 min and 2–32 h) and were analysed for their fluoroquinolone concentrations using a competitive enzyme immunoassay. The metabolization of enrofloxacin to ciprofloxacin was determined using liquid chromatography-mass spectrometry. The maximum serum concentrations of fluoroquinolones were observed at the time point of 2.82 ± 0.1 h and amounted to 10.67 ± 0.5 µg/mL. Fluoroquinolones redistributed to a minor extent into the aqueous humour reaching maximum concentrations of 4.52 ± 1.2 µg/mL after 7.54 ± 1.0 h of drug administration. The mean residence time (MRT), volume of distribution (V_d_), and terminal half-life (t_1/2_ ß) were 1.68-, 2.84-, and 2.01-fold higher in aqueous humour than in serum, indicating that fluoroquinolones were trapped in aqueous humour. Enrofloxacin was only marginally metabolized into ciprofloxacin. A single intramuscular injection of a therapeutical dose of enrofloxacin (25 mg/kg BW) thus generated sustained and therapeutically active levels of enrofloxacin in the aqueous humour of chicken eyes.

## 1. Introduction

Ocular diseases in birds frequently result from systemic infections. Many bacteria are known to induce endophthalmitis with uveitis, iris swelling, synechia, and other pathological alterations of the eye [1,2]. *Salmonella* spp. and *Mycoplasma* spp. are frequently involved; in addition, *Pasteurella multocida*, *Pseudomonas aeruginosa*, *Escherichia coli*, *Staphylococcus* spp., *Micrococcus* spp., *Bordetella avium*, *Erysipelotrix rhusiopathiae*, and *Listeria monocytogenes* are also known for producing eye diseases and pathologies of the anterior eye chamber [1,3,4]. As vision is of fundamental importance for birds, assuring a fully functional sense of visual perception by initiating rapid and effective therapy may be vital to the avian patient [5].

Antibiotic treatment of intraocular infections is generally difficult, since the blood–ocular barrier, a natural mechanism of protection of the eye, impedes the intraocular penetration and action of many drugs [6].

A group of antibiotics commonly used in human ophthalmology comprises the fluoroquinolones [7]. Because of its favourable characteristics, enrofloxacin, a fluoroquinolone antimicrobial agent, is frequently used in avian medicine for the treatment of many types of systemic bacterial infections. It is bactericidal at low concentrations against a broad spectrum of Gram-negative and Gram-positive bacteria, as well as *Mycoplasma* or *Mycoplasmopsis* spp. and *Chlamydia* spp. [8,9]. Furthermore, enrofloxacin is well tolerated in birds. Even at high concentrations over a long treatment period, only few side effects are observed [9]. Finally, it shows favourable pharmacokinetic properties, such as rapid absorption at the gastro-intestinal tract, resulting in high blood and tissue concentrations, even in bones and the central nervous system [10,11,12]. There is a growing need for the prudent use of antibiotics in order to guarantee efficacy and to prevent the development and spread of antimicrobial resistance [13]. Several studies revealed the suitability of enrofloxacin in veterinary ophthalmology and its ability to penetrate the blood–ocular barrier in mammals. Enrofloxacin concentrations above the minimal inhibitory concentrations (MICs) and the minimal bactericidal concentrations (MBCs) of *Leptospira* were obtained in the vitreous humour of horses when treating intraocular leptospirosis [14]. In addition, therapeutic concentrations of enrofloxacin in the aqueous humour of dogs were measured after a single subcutaneous application [15]. Corresponding studies in avian medicine are lacking so far, although fast and effective treatment of intraocular bacterial infections is of exceptionally high relevance in birds. However, a direct transfer of pharmacokinetic parameters of enrofloxacin from mammals to birds is not possible because of the bird-specific anatomical and physiological particularities of the blood–ocular barrier.

The anatomy and physiology of the avian eye substantially differ from those in mammals in several aspects, such as the blood supply. In mammals, the blood–ocular barrier system is formed by two main components: the blood–aqueous barrier, comprising the epithelial cells of the processus ciliaris majoris of the corpus ciliare, as well as the blood–retinal barrier, consisting of tight junctions between retinal capillary endothelial cells and retinal pigment epithelial cells. Both barriers provide a suitable, highly regulated chemical environment for the avascular transparent tissues of the eye and are a drainage route for the waste products of the ocular metabolism [6,16]. The anatomy of the blood–aqueous barrier of birds is similar to that of mammals. Aqueous humour is produced continuously by the inner epithelial layer of the ciliary processes of the ciliar body and is secreted into the posterior chamber (camera posterior bulbi). It further flows through the pupillary opening into the anterior chamber (camera anterior bulbi), where it is finally drained through the spaces of Fontana (spatia anguli iridocornialis) into the scleral venous sinus (sinus venosus sclerae) [17,18].

The main anatomical and physiological differences between birds and mammals refer to the second part, the blood–retinal barrier. In contrast to mammals, the avian retina is avascular. It is assumed that the maintenance of the intraocular milieu and the nutritional supply are mainly carried out by the capillary network of the choroid (lamina choriocapillaris) and the richly vascularised pecten oculi, a pleated, vaned, or conical projection of the retina, which is an anatomical structure unique to the avian eye and similar to the conus papillaris of reptiles. Despite extensive investigations, the function of the pecten oculi is in part unclear, but the characteristics of the capillary endothelium in the pecten are suggestive of a constant active transepithelial transport. The pecten oculi thus probably forms an important component of the avian blood–retinal barrier [18,19].

The aim of this study was to investigate whether an intramuscular injection of enrofloxacin at a dosage of 25 mg/kg BW can achieve clinically effective concentrations of enrofloxacin in the aqueous humour of clinically healthy chickens. The results provide a scientific basis for the appropriate systemic treatment of intraocular infections with sensitive bacteria in avian patients. As enrofloxacin is an antibiotic of critical importance in veterinary medicine, the present study aimed to contribute to a responsible and targeted use of enrofloxacin in avian medicine.

## 2. Materials and Methods

### 2.1. Animals

The experiments were conducted in accordance with German animal welfare regulations and with permission from the German authorities (reference number ROB 55.2-2532.Vet_02-19-165). The study included 40 adult, clinically healthy roosters (*Gallus gallus domesticus*, Lohman Selected Leghorn). All birds were hatched and raised at Clinic for Birds, Small Mammals, Reptiles and Ornamental Fish of Ludwig Maximilians University, Munich, and were of the same age and genetic background. Animals were housed in two groups of 20 animals each under identical conditions. Food and drinking water were supplied ad libitum. Animals were drug naïve for at least two months prior to experimentation.

The chickens were determined to be clinically healthy based on visual inspections during daily care and a complete physical and ophthalmological examination before administration of enrofloxacin and sampling. The average body weight was 2.39 +/− 0.19 kg at the beginning of the first study cycle and 2.31 +/− 0.22 kg in the second cycle.

### 2.2. Dosing and Sample Collection Protocols

Animals were treated with a single intramuscular injection of enrofloxacin (Baytril^®^; 50 mg/mL; Bayer HealthCare AG, Leverkusen, Germany) in the deep pectoral muscle (musculus supracoracoideus) at a dosage of 25 mg/kg body weight (BW). The experiments were conducted in two cycles.

In cycle 1, aqueous humour from the left eye and blood from the right jugular vein were collected at the time points of 2, 4, 8, 15, 30, and 60 min after administration of enrofloxacin. After a wash-out period of 3 weeks, the experiment was repeated for sampling time points of 2, 4, 8, 16, and 32 h using the right eye instead of the left one (cycle 2). Untreated animals served as controls (time point 0). For each time point, 6 individual animals were randomly assigned from the study population. Using this experimental setup, each animal was sampled once in cycles 1 and 2 in order to avoid multiple corneal punctures of the same eye in short intervals.

To further reduce stress and the risk of injury during sampling, manipulations were performed under general anaesthesia using isoflurane (initiating concentration of 5%, maintenance at 2–2.5%) starting approximately 5 min before sampling. Because of practical reasons, the roosters used for 2 and 4 min sampling were sedated with anaesthesia before enrofloxacin injection. For additional local corneal analgesia, oxybuprocaine hydrochloride (Conjucain^®^ EDO^®^; 4.0 mg/mL; eye drops; Bausch & Lomb, Rochester, NY, USA) was applied to the conjunctival sack 1 min before sampling. The collection of aqueous humour was performed by means of paracentesis. The anterior chamber of the eye was punctured using an insulin syringe slightly dorsally of the temporal eye corner. A volume of 0.05 mL of aqueous humour was aspirated and subsequently replaced with the same volume of sterile isotonic saline solution to maintain intraocular pressure. For the determination of the serum levels, blood samples of 1 mL were collected from the right jugular vein, using a 22-gauge needle and a 2 mL syringe.

All chickens were re-examined 20 min after manipulation and for consecutive days by means of visual inspection during daily care. None of the roosters showed signs of pain, eye disorders, nor visual impairment as a consequence of paracentesis.

In cycle 2, animals were euthanized immediately after sampling with the injection of pentobarbital sodium (Release^®^; 300 mg/mL; WDT, Garbsen, Germany) at a dosage of 200 mg/kg [20] injected into the right jugular vein.

Samples of aqueous humour were stored at −18 °C until analysis. Blood samples were collected in serum tubes and allowed to coagulate for at least 30 min at room temperature, before serum was collected after centrifugation at 3500× *g* for 3 min and stored at −18 °C until use.

### 2.3. Determination of Fluoroquinolones

Fluroquinolone concentrations in serum and aqueous humour were determined using a generic enzyme immunoassay (EIA) with a highly sensitive monoclonal antibody as described in detail before [21]. The EIA was originally developed for the detection of fluoroquinolone residuals in milk and foodstuff. Its usefulness for serum and aqueous humour was confirmed by spiking serum and aqueous humour samples with known concentrations of enrofloxacin, resulting in recovery rates of >90%.

Microtiter plates were first coated with rabbit anti-mouse antibodies (5 µg/mL; Z 0259; DAKO, Hamburg, Germany; 100 µL/well) in carbonate buffer (0.05 M; pH 9.6) overnight at room temperature and stored at 4 °C for up to 3 weeks until use. Free binding sites were blocked with 3% (*w/v*) sodium caseinate (150 µL/well) in phosphate-buffered saline (PBS; 0.01 M phosphate buffer with 0.1 M NaCl; pH 7.3) for 30 min. Microtiter plates were washed 3 times with 146 mM NaCl solution containing 0.025% (*v/v*) Tween 20, before plates were incubated with 2 ng/well mouse anti-norfloxacin monoclonal antibody 5H8 [21] in phosphate-buffered saline for 60 min. After washing the plates again four times, enrofloxacin standards or appropriate dilutions of the samples (serum or aqueous humour diluted in PBS containing 0.1% bovine serum albumin) were added together with a clinafloxacin-horseradish peroxidase enzyme conjugate (50 µL each/well) and incubated for 60 min. Plates were washed again and enzyme reaction was developed with the addition of 100 µL of substrate solution [21]. After incubation in the dark for 20 min, enzyme reaction was stopped by adding 1 M sulfuric acid (H_2_SO_4_; 100 µL/well) and absorbance was measured at λ = 450 nm with the reference wavelength of 620 nm.

### 2.4. Determination of Enrofloxacin/Ciprofloxacin Ratios

The EIA is group specific but does not differentiate between individual fluoroquinolones. For this, the enrofloxacin/ciprofloxacin ratio was determined in three serum and aqueous humour samples using liquid chromatography with tandem mass spectrometry (LC-MS/MS).

Enrofloxacin and ciprofloxacin were extracted stepwise from serum with acetonitrile (2 × 12.5 mL) and citrate-phosphate buffer (pH 6.0; 0.1 M citric acid, 0.1 M Na_2_HPO_4_, and 2 mM aqueous EDTA solution; 5 mL). Following a centrifugation step, an aliquot of 15 mL of the supernatant was transferred into a tube containing 1.0 g of sodium chloride. A volume of 3.3 mL of ethyl acetate was added, and the tube was shaken. After centrifugation, the organic phase was transferred into another tube and ethylene glycol (ethylene glycol/acetonitrile 1 + 3 (*v/v*); 0.8 mL) was added as keeper. The solvent was evaporated; the residue was dissolved in water; and the final extract was analysed with LC-MS/MS. The described method has been validated for the determination of antibiotics in muscle, kidney, liver, milk, egg, and cheese using LC-MS/MS according to Commission Decision 2002/657/EC and was adapted for the matrix serum.

From aqueous humour, enrofloxacin and ciprofloxacin were isolated using immunoaffinity chromatography. Columns were prepared by conjugating a generic anti-quinolone monoclonal antibody [21] to CNBr activated sepharose 4B (GE Healthcare, Munich, Germany). PBS was used for washing. Analytes were eluted using pure methanol and directly used for enrofloxacin and ciprofloxacin determination using LC-MS/MS.

### 2.5. Data Analysis

Pharmacokinetic parameters for overall fluoroquinolones were calculated with a non-compartmental analysis using PK Solutions 2.0 computer software (Summit Research Services, Montrose, CO, USA) from the mean values ± SDs of *n* = 6 individual observations. Serum data were analysed using a three-exponential model after i.m. administration of a single dose. Aqueous humour data best fitted a two-exponential model. The variables calculated from the curves using the regression analysis included the values for maximum serum concentration (C_max_; µL/mL), time to maximum serum concentration (T_max_; h), mean resident time (MRT; h), volume of distribution (V_d_; L/kg) and renal clearance (Cl; L/h/kg), and terminal half-life (t_1/2_ ß; min). The area under the curve from T0 to infinity (AUC_0–inf_) was calculated using the linear trapezoidal rule. The fraction of fluoroquinolones redistributed from serum to aqueous humour as well as the overall fraction of enrofloxacin metabolized to ciprofloxacin were calculated using the AUC (0-inf).

Statistical analyses were carried out when appropriate using unpaired *t*-tests included in Graphpad Prism Version 9 (San Diego, CA, USA).

### 2.6. Minimal Inhibitory Concentrations (MICs) of Enrofloxacin

To find relevant and substantiated data on the MIC values of fluoroquinolones against bacteria significant in avian ophthalmology, a standardized literature search in PubMed covering the publication interval from January 2015 to September 2022 was performed using the keywords birds, enrofloxacin, MIC, eye, ophthalmology, and ophthalmological, as well as the name of the respective pathogen.

## 3. Results

### 3.1. Tolerability of Enrofloxacin

No adverse effects to enrofloxacin administered as a single intramuscular dose of 25 mg/kg BW were observed in the chickens. All birds showed normal behaviour, and feed and water intake after treatment. Potential side effects described in the literature, such as increased water intake and polyuria, apathy, reduced feed intake, vomiting, and diarrhoea, or central nervous disorders were not detected during the study.

### 3.2. Pharmacokinetics of Fluoroquinolones in Serum and Aqueous Humour

The time course of fluoroquinolone concentrations in serum and aqueous humour detected using the EIA is shown in Figure 1.

After a single intramuscular dose of 25 mg/kg enrofloxacin, maximum serum concentrations were observed at the time point of 2.82 ± 0.1 h and amounted to 10.67 ± 0.5 µg/mL. Fluoroquinolones redistributed to a minor extent into the aqueous humour, reaching maximum concentrations of 4.52 ± 1.2 µg/mL after 7.54 ± 1.0 h of drug administration. This represents a fraction of 64.07% of fluoroquinolones penetrating from serum into the aqueous humour as calculated using AUC_0-inf_ values. The mean residence time (MRT), volume of distribution (V_d_), and terminal half-life (t_1/2_ ß) were 1.68-, 2.84-, and 2.01-fold higher in aqueous humour than in serum, indicating that fluoroquinolones were trapped in aqueous humour. All pharmacokinetic parameters are summarized in Table 1.

### 3.3. Metabolization of Enrofloxacin to Ciprofloxacin

The fluoroquinolone EIA is not specific to enrofloxacin but also detects other fluoroquinolones, including ciprofloxacin, the main metabolite of enrofloxacin. Therefore, to determine to which proportion enrofloxacin was metabolized to ciprofloxacin, additional LC-MS/MS analyses were performed for selected serum and aqueous humour samples.

Ciprofloxacin concentrations in serum were 0.17, 0.72, and 0.71 µg/mL at the time points of 1, 4, and 8 h. The corresponding enrofloxacin concentrations were 8.74, 6.53, and 4.93 µg/mL. This represents ciprofloxacin proportions of 1.9%, 9.9%, and 12.6% of the overall fluoroquinolone concentrations as found with the EIA.

Ciprofloxacin concentrations in aqueous humour were 0.02, 0.05, and 0.08 µg/mL at the time of points 1, 4, and 8 h. The corresponding enrofloxacin concentrations were 1.3, 1.46, and 1.48 µg/mL, indicating that ciprofloxacin penetrated into the aqueous humour to a smaller extent than enrofloxacin (1.4%, 3.3%, and 5.4% of total fluoroquinolone EIA reactivity).

The individual data are given in Table 2.

### 3.4. Fluoroquinolone Susceptibility to Relevant Bacteria

In order to evaluate the potential therapeutical applicability, the measured fluoroquinolone concentrations in aqueous humour (reaching a maximum of 4.52 µg/mL) were compared with the minimal inhibitory concentrations (MICs) of some relevant, enrofloxacin-sensitive bacteria. Table 3 shows a summary of pathogens, for which more recent data could be retrieved from PubMed. No literature related to both MIC values and avian ophthalmology was available. The evaluated data all refer to poultry.

## 4. Discussion

The present study demonstrates that a single intramuscular injection of a therapeutical dose of enrofloxacin (25 mg/kg BW) generates sustained and high levels of enrofloxacin in aqueous humour. The concentrations detected were well above the MICs of clinically relevant bacteria causing eye infections. As shown by the LC-MS/MS results, enrofloxacin is only marginally metabolized into ciprofloxacin in chickens and thus represents the active compound in aqueous humour. Due to the long terminal half-life of enrofloxacin in aqueous humour of 8.79 h, a single daily dose of enrofloxacin is sufficient to produce sustained and therapeutically effective drug concentrations in the eye.

### 4.1. Study Design

The chicken eye represents a frequently used model system in ophthalmology [30]. Pharmacokinetic studies usually employ repeated sample collection at different time points after drug administration and analysis of individual time vs. concentration curves. Due to the minimum volume required for laboratory analyses, we decided to perform a randomized study using 6 individual animals out of a pool of 40. This allowed us to sample aqueous humour from intact eyes at each time point to exclude artifacts due to prior paracenteses. Because of animal welfare reasons, we decided to perform paracentesis using general anaesthesia despite the fact that this could affect the outcome of drug delivery. General anaesthesia generally causes a drop in heart rate and blood pressure. It could be thus deduced that drug concentrations might be reduced in serum and aqueous humour of anaesthetized compared with unanaesthetized birds. The results of our study might thus even underestimate the true efficacy of the drug.

### 4.2. Fluoroquinolone Analysis

A generic enzyme immunoassay (EIA) with a highly sensitive monoclonal antibody against fluoroquinolones was used to measure the fluoroquinolone concentrations in serum and aqueous humour samples. Compared with other standard methods, the EIA is a cheap, fast, and easy-to-learn analytical method. The assay system used in this study has a detection limit of 0.17 ng/mL for enrofloxacin, as previously determined [21], and thus belongs to the most sensitive EIAs described so far for the detection of fluoroquinolones. In comparison, the limits of detection of enrofloxacin in milk with liquid chromatography and mass spectrometry (LC-MS/MS) range from 0.25 to 1.5 ng/mL [31,32]. The used EIA does not distinguish among different fluoroquinolones, such as enrofloxacin and its main metabolite, ciprofloxacin. The results, therefore, reflect the total amount of fluoroquinolones contained in each sample. Some subsamples of aqueous humour and serum were, therefore, additionally investigated using LC-MS/MS, confirming that enrofloxacin is only marginally metabolized into ciprofloxacin in chickens [33] and represents the main active compound in serum as well as in aqueous humour.

### 4.3. Pharmacokinetics

To be effective against clinically relevant intraocular bacterial infections, enrofloxacin must achieve concentrations above the MICs for the time required by the individual pharmacodynamic properties of the causative bacteria [34].

Superficial eye infections can be effectively treated with topical antibiotics. The locally effective concentration of the antibiotic can be easily maintained with frequent applications, and prior determination of the causative germs enables the targeted selection of a suitable preparation. However, the topical application of ciprofloxacin and newer fluoroquinolones failed to produce clinically relevant concentrations in the aqueous humour of humans [35] and horses [36,37]. Thus, the antimicrobial treatment of bacterial infections of the interior eye requires systemic treatment either alone or in combination with topical preparations. Due to the selectivity of the blood–aqueous barrier, which represents a special protective function in the eye, there are only a few antimicrobials that are suitable for this purpose [38]. Low-fat-soluble antibiotics such as penicillins, cephalosporins, and aminoglycosides are hardly able to pass the intact barrier and reach only marginal concentrations in the posterior eye region [39].

Fluoroquinolones are drugs of choice in the systemic treatment of intraocular infections in humans. Especially ciprofloxacin and its more lipophilic derivatives, ofloxacin, moxifloxacin, and others, have been evaluated in humans [7,40]. Because these are not available for veterinary medicine, the present study evaluated the ability of enrofloxacin to fulfil the requirements for the treatment of intraocular infections. For this, the pharmacokinetics of a single intramuscular injection of 25 mg/kg enrofloxacin were determined simultaneously in serum and aqueous humour over time. The study extends our knowledge on the pharmacokinetic parameters of enrofloxacin after intramuscular administration and shows that an increase in dose from 5 to 25 mg/kg results in an equivalent increase in maximal serum concentrations (2.1 vs. 10.67 µg/mL [41]). Dose- and route-of-application-independent parameters such as terminal half-life, MRT, Vd, and Cl were in the range of published data, demonstrating the suitability of the EIA for the reliable and sensitive measurement of fluoroquinolones in serum.

In Germany, enrofloxacin is currently approved for the treatment of digestive and respiratory diseases in companion birds and poultry caused by a variety of bacterial species, with a dosage of 10 mg/kg body weight for poultry and a dosage of 20 mg/kg body weight for ornamental birds. The approval further refers to the possible need to increase the dose in case of complicated infections. In addition, dosing regimens for a variety of indications, ranging from 5 mg/kg BW to 30 mg/kg BW once or twice daily, can be found in the literature [42,43,44]. Due to the lower enrofloxacin levels in aqueous humour compared with serum, off-label use at a concentration of 25 mg/kg BW appears suitable for the treatment of ophthalmic infections in birds, an off-label indication in Germany.

Our data also verify that enrofloxacin is only marginally metabolized into ciprofloxacin in chickens. At the time point of 8 h after treatment, only 12.6% of the original enrofloxacin dose had been converted into ciprofloxacin. This value is somewhat lower than the 26% metabolization rate reported for 9 d old broiler chickens orally treated for 5 days at a dose of 15.5 mg/kg/d [33]. Because the proportion of ciprofloxacin was smaller in aqueous humour than in serum (5.4 vs. 12.6%), our results indicate that enrofloxacin represents the main pharmacologically effective substance in the eye after parenteral application in chickens.

The enrofloxacin levels in aqueous humour amounted to about 64% of serum concentrations and reached a peak of 4.52 ± 1.2 µg/mL after 7.54 h of treatment. Enrofloxacin levels remained high, reaching 3.47 ± 1.26 µg/mL after 16 h. Together with the calculated terminal half-life of 8.79 ± 0.93 h and the mean residence time of 16.03 ± 1.09 h, these results demonstrate that a single dose of enrofloxacin is sufficient to establish continuously high drug levels in the aqueous eye compartment.

The drug concentrations in aqueous humour observed in chickens were dramatically higher than those determined for enrofloxacin in horses [45]. In that study, horses were given enrofloxacin at a dose of 7.5 mg/kg body weight once daily intravenously for 4 consecutive days. Enrofloxacin concentrations were determined in aqueous humour 1 h after the last injection. The results showed that only a fraction of 12.1% of enrofloxacin was transferred into the eye, reaching a mean concentration of 0.54 µg/mL in aqueous humour. This is about 16-fold lower than that observed in the present study. The concentrations of enrofloxacin observed in chickens are also a magnitude higher than those determined for ciprofloxacin after oral administration in humans [46]. These data suggest that enrofloxacin penetrates to a higher extent into the aqueous humour of chickens than in mammals due to anatomical and functional differences in the blood–retinal barrier. However, to confirm this assumption, corresponding dosages have to be tested in mammals in future investigations.

Inflammatory changes in the ocular structures caused by infection might change drug penetration into the eye. Especially, diseases of the uvea have been described to cause a breakdown of the blood–ocular barrier system, which could further facilitate drug entry [47]. Consequently, intraocular infections might even increase enrofloxacin penetration in aqueous humour. However, this hypothesis needs to be confirmed with further experiments, as our study included exclusively healthy chickens in order to first obtain information about the usability of enrofloxacin in the treatment of intraocular bacterial infections.

## 5. Conclusions

The results of the present study demonstrate that enrofloxacin readily penetrates into the aqueous humour of the eye after systemic administration of a single enrofloxacin dose. Based on pharmacokinetic/pharmacodynamic considerations, enrofloxacin appears to be suitable for the treatment of intraocular infections in chickens with strains of Gram-negative bacteria having MIC values of ≤1.

## Figures and Tables

**Figure 1 vetsci-10-00005-f001:**
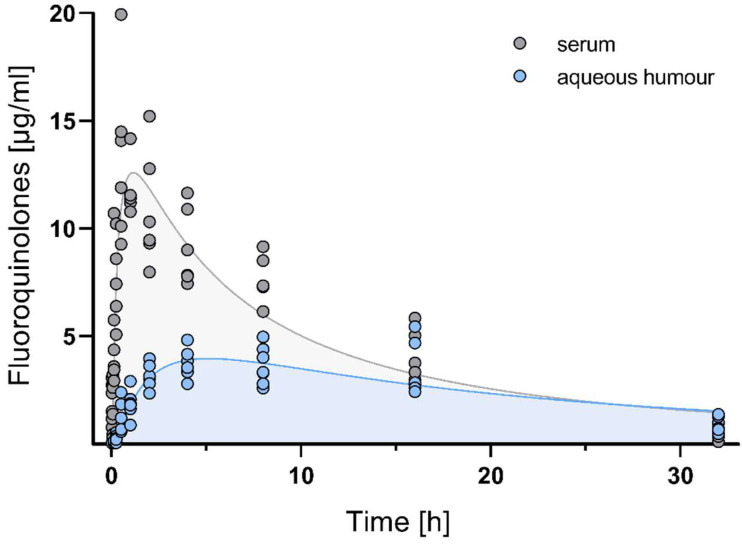
Time course of fluoroquinolone EIA reactivity in serum and aqueous humour after a single intramuscular injection of 25 mg/kg enrofloxacin in chickens. Each circle represents the measured serum concentration of an individual out of *n* = 6 animals per time-point.

**Table 1 vetsci-10-00005-t001:** Pharmacokinetic parameters for fluoroquinolones in serum and aqueous humour after a single intramuscular application of 25 mg/kg body weight.

	Serum		Aqueous Humour	
Parameter	Mean ± SD	Range	Mean ± SD	Range
C_max_ (µg/mL)	10.67 ± 0.5	10.2–11.3	4.52 ± 1.2	3.2–6.6
t_max_ (h)	2.82 ± 0.1	2.7–2.9	7.54 ± 1.0	6.2–8.7
AUC_0-inf_ (µg·h/mL)	155.40 ± 39.1	117.58–204.75	99.57 ± 26.8	71.88–137.83
MRT (h)	9.52 ± 1.2	8.18–11.28	16.03 ± 1.9 *	12.84–17.89
V_d_ (L/kg)	1.11 ± 0.2	0.88–1.46	3.16 ± 0.91 *	1.97–4.2
Cl (L/h/kg)	0.17 ± 0.04	0.18–0.21	0.27 ± 0.06 *	0.18–0.35
t_1/2_ ß (h)	4.36 ± 0.38	4.07–5.06	8.79 ± 0.93 *	7.54–10.37

* Significantly different from serum at *p* < 0.001.

**Table 2 vetsci-10-00005-t002:** Enrofloxacin (ENR) and ciprofloxacin (CPR) concentrations in serum and aqueous humour as measured using LC-MS/MS and the corresponding proportions of ciprofloxacin in the total amount of fluoroquinolones.

	Serum			Aqueous Humour		
Time Point	ENR (µg/mL)	CPR (µg/mL)	Proportion of CPR (%)	ENR (µg/mL)	CPR (µg/mL)	Proportion of CPR (%)
1 h	8.74	0.17	1.9	1.3	0.02	1.4
4 h	6.53	0.72	9.9	1.46	0.05	3.3
8 h	4.93	0.71	12.6	1.48	0.08	5.4

**Table 3 vetsci-10-00005-t003:** Minimal inhibitory concentrations (MICs) of enrofloxacin-sensitive bacteria in birds. Data are based on papers found in PubMed using the keywords birds, enrofloxacin, MIC, eye, ophthalmology, and ophthalmological, and the name of the respective pathogen published between January 2015 and September 2022.

Bacteria	MIC	Avian Host	Geographic Area	References
*Salmonella enteritidis*	0.0625–1 µg/mL	Broiler	South Korea	[22]
*Salmonella typhimurium*	0.12–16 µg/mL	Chicken	China	[23]
*Mycoplasmopsis* *(Mycoplasma) synoviae*	4–32 µg/mL	Chicken	China	[24]
	0.031–32 µg/mL	Poultry	Europe	[25]
	0.625–>10 µg/mL	Chicken	Asia	[26]
	1.0–>16 µg/mL	Poultry	Italy	[27]
*Mycoplasma gallisepticum*	0.031–16 µg/mL	Poultry	Europe	[25]
	≤0.039–5 µg/mL	Chicken	Asia	[26]
*Escherichia coli*	8–256 µg/mL	Chicken	Asia	[23]
	0.5 µg/mL	Chicken	Asia	[28]
	0.016–>16 µg/mL	Poultry	Germany	[29]

## Data Availability

Not applicable.

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
