# Peer review of "Penetration of Enrofloxacin in Aqueous Humour of Avian Eyes"

_vetsci, 2022, doi:10.3390/vetsci10010005_

Round 1
Reviewer 1 Report
Vet. Sci. 2022
The manuscript is well written, and a wide range of factors have been considered to study the distribution of enrofloxacin in aqueous humour of avian eyes.
To further improve the quality of the work, I suggest the following:
A) The authors in the conclusions (lines: 376-379) write that “The results of the present study demonstrate that enrofloxacin penetrates readily into the aqueous humour of the eye after systemic administration of a single enrofloxacin dose reaching therapeutically effective concentrations above the MIC of clinically relevant intraocular infectious bacteria.”
The authors must be rewriting the conclusion. As they know it is not enough only the concentration of the drug to be above the MIC. For the fluoroquinolones, drugs that are time and dose dependent antimicrobials, the effectiveness dependent for the ratio of AUC/MIC ≥ 125 (for gram – bacteria) or Cmax/MIC ≥ 10.
From the results of the study the AUC0-inf (μg·h/ml) of enrofloxacin in aqueous humour are 99.57 ± 26.8 (Mean ± SD) and the range are 71.88 - 137.83. If we take in to consideration the range of MIC of Escherichia coli, only the strains who have MIC ≤1 or 0,5 μg/mL are sensitive to enrofloxacin -AUC/MIC ≥ 125 (as an example).
B) The authors should explain why they did not measurement the drugs concentration (enrofloxacin and its main metabolite ciprofloxacin) in all blood and aqueous humour samples by LC-MS/MS. The method that is more accurate and sensitive. If the use the LC-MS/MS method may be the metabolism of enrofloxacin into ciprofloxacin was different.
Author Response
The manuscript is well written, and a wide range of factors have been considered to study the distribution of enrofloxacin in aqueous humour of avian eyes.
To further improve the quality of the work, I suggest the following:
- A) The authors in the conclusions (lines: 376-379) write that “The results of the present study demonstrate that enrofloxacin penetrates readily into the aqueous humour of the eye after systemic administration of a single enrofloxacin dose reaching therapeutically effective concentrations above the MIC of clinically relevant intraocular infectious bacteria.”
The authors must be rewriting the conclusion. As they know it is not enough only the concentration of the drug to be above the MIC. For the fluoroquinolones, drugs that are time and dose dependent antimicrobials, the effectiveness dependent for the ratio of AUC/MIC ≥ 125 (for gram – bacteria) or Cmax/MIC ≥ 10.
From the results of the study the AUC0-inf (μg·h/ml) of enrofloxacin in aqueous humour are 99.57 ± 26.8 (Mean ± SD) and the range are 71.88 - 137.83. If we take in to consideration the range of MIC of Escherichia coli, only the strains who have MIC ≤1 or 0,5 μg/mL are sensitive to enrofloxacin -AUC/MIC ≥ 125 (as an example).
Thank you very much for this important point. We now have incorporated this issue and rewritten the conclusions section accordingly (Lines 392-396).
PK/PD-modelling greatly contributes to our understanding of the effectiveness of antibiotics and is an important discipline, but only infection experiments could really clarify to what extent effectiveness is given. However, this went beyond the aim of the current study.
- B) The authors should explain why they did not measurement the drugs concentration (enrofloxacin and its main metabolite ciprofloxacin) in all blood and aqueous humour samples by LC-MS/MS. The method that is more accurate and sensitive. If the use the LC-MS/MS method may be the metabolism of enrofloxacin into ciprofloxacin was different.
The EIA system was choosen for two reasons: First, the described EIA is more sensitive than the LC-MS/MS method. As mentioned in the manuscript, the limit of detection of enrofloxacin is 0.17 ng/ml vs. 0,25 and 1,5 ng/ml with liquid chromatography and LC-MS/MS). Second, the EIA enables measurement of biological samples without prior extraction. This is of particular advantage when small sample volumes have to be measured as in our study (maximum volume of 0.05 ml aequous humour per chicken eye). Due to the small sample volume we analyzed only a limited number of samples of selected time points to evaluate the fraction of enrofloxacin metabolized to ciprofloxacin.
We want to thank the reveiwer for the careful review and the helpful comments.
Reviewer 2 Report
This manuscript aimed to investigate the clinical efficacy of enrofloxacin in bird eyes. The methods, number of included birds, results and discussion appear appropriate.
The manuscript also is very well written.
Since the broad aim of this manuscript to test enrofloxacin efficacy in treatment of the treatment of intraocular infections in birds, I was expecting to include diseased bird in the study design. However, the authors only tested clinically healthy birds. How could the degree of infections could affect the drug penetration ?
What is the basis of choosing this specific dose of administration?
Did the authors evaluate different dose and different route?
Did the author also evaluate the withdrawal time of the enrofloxacin from the birds?
The discussion section is also very well written based on the collected data.
I think acknowledging these comments, I believes the manuscript would be suitable for publications after minor revision.
Author Response
This manuscript aimed to investigate the clinical efficacy of enrofloxacin in bird eyes. The methods, number of included birds, results and discussion appear appropriate.
The manuscript also is very well written.
Since the broad aim of this manuscript to test enrofloxacin efficacy in treatment of the treatment of intraocular infections in birds, I was expecting to include diseased bird in the study design. However, the authors only tested clinically healthy birds. How could the degree of infections could affect the drug penetration?
This is indeed an interesting issue. Pharmacokinetic studies are usually done in healthy animals to assess the principle pharmacokinetic parameters. These may change in diseased states when for example biological barriers are affected. Because inflammation due to bacterial infection is expected to affect barrier function, permeability of drugs into the aqueous humour should be increased. The extent of drug penetration into the aqueous humour in the presence of intraocular infection is usually evaluated in population kinetics studies on a larger number of patients. However, such investigations would exceed the scope of the present study which was only intended to demonstrate the suitability of enrofloxacin to treat bacterial eye infections in birds. This issue is now stated in the discussions section (Lines 382-389).
What is the basis of choosing this specific dose of administration?
The study was conducted on chickens as a pharmacological model. However, the results of the study are expected to benefit primarily ornamental birds and birds of prey. In Germany enrofloxacin is currently approved for poultry at an oral dose of 10 mg/kg body weight. In ornamental birds, Baytril 25mg/ml Injection is labeled for treatment of infections of the gastrointestinal and respiratory tract at a dose of 20 mg/kg body weight i.m. once daily. Because of the protective mechanisms of the eye which might impede drug penetration, we decided to use a slightly higher dose. The requirement to increase the dose is confirmed by our results, since the measured enrofloxacin concentrations in aqueous humour revealed significantly lower drug concentrations than in serum. It is therefore not certain whether therapeutically effective concentrations would still be ensured at significantly lower doses. In the current literature, dosages of up to 30 mg/kg BW twice daily are recommended for ornamental birds, depending on the infectious agent and site of infection (Hawkins et al: Birds. In Exotic Animal Formulary, 5th ed.; Carpenter, J.W., Marion, C.J., Eds.; Elsevier Health Sciences: 2018; pp. 178-179., Doneley, B. Formulary. In Avian Medicine and Surgery in Practice: Companion and Aviary Birds, Second ed; Doneley, B., Eds.; CRC Press, Taylor and Francis Group: New York, 2016: pp. 425-454.). The chosen dosage of 25 mg/kg once daily is within the suggested range and not endowed with side effects.
Based on the reviewer’s comments, we clarified this point in our paper, so we made some additions (Lines 348-354).
Did the authors evaluate different dose and different route?
We haven’t performed dose-finding studies or evaluated different routes of application. This will be subject of clinical studies required for market authorization. From the pharmacokinetic data published, it appears conceivable that oral application of enrofloxacin at tolerable doses would result in therapeutically active drug levels in serum and aqueous humour.
Did the author also evaluate the withdrawal time of the enrofloxacin from the birds?
This is also an interesting point but was not investigated in the context of the present study, because we used the chicken as a model system for ornamental birds. In addition, depletion studies are usually done with twice the dose as used for treatment. From this, the experiments done in the present study would have been repeated with a higher dose.
The discussion section is also very well written based on the collected data.
I think acknowledging these comments, I believes the manuscript would be suitable for publications after minor revision.
We would like to thank the reviewer for the careful review and the helpful comments to further improve the manuscript.
Reviewer 3 Report
Overall Comments
In this study, the Authors determined the pharmacokinetic profile in serum and aqueous humour of enrofloxacin following a single intramuscular administration in chicken at the dosage of 25 mg/kg BW. Specifically, the Authors investigated the feasibility of this regimen to achieve clinically effective concentrations for the possible treatment of intraocular infections with sensitive bacteria.
The manuscript is well-written, with a proper experimental study design to provide pharmacokinetic information but not proper to provide therapeutic indications.
The novelty is clear, but the importance of this investigation due to the implications within the topic of antimicrobial stewardship needs to be further discussed.
To be this manuscript considered for publication, the Reviewer needs some major concerns to be sincerely addressed by the Authors.
Simple Summary
Lines 23-24: Please, remove this sentence or change it since the study was not performed on animals affected by eye infections, so is not possible to state that enrofloxacin is useful for the treatment of these conditions.
Introduction
Lines 59-61 and 67-68: “Because of its favourable characteristics, enrofloxacin, a fluoroquinolone antimicrobial agent, is frequently used in avian medicine”.
Do the Authors think using enrofloxacin for many types of systemic bacterial infection can be considered a prudent antimicrobial use? Enrofloxacin is a human and veterinary critically important antimicrobial, and its use should be limited to infections sustained by bacteria resistant to first- and second-choice active compounds.
Materials and Methods
Line 127: How did the Authors choose the dosage? Why was it not decided to have two treatment groups, one with a registered dosage and one off-label?
Line 137-141: Were the Authors not concerned that alterations in the cardiovascular pattern related to general anesthesia, particularly systemic pressure, might affect the outcome of drug delivery?
Lines 153-155: Out of curiosity: since the animals were to be euthanized, did the Authors also consider conducting a residual depletion study?
Lines 198-199: Why did the Authors choose two different extraction methods for the two matrices (i.e. serum and aqueous humour)?
Discussion
Lines 326-328: If these antimicrobial “are hardly able to pass the intact barrier”, why did the Authors choose an experimental model with only healthy animals? Instead of a critically important antimicrobial within an experimental animal model it was possible to try other active compounds in sick animals with an altered blood-retinal barrier.
Lines 329-330: “Fluoroquinolones are drugs of choice in systemic treatment of intraocular infections 329 in man”. Exactly, and in particular for this reason it is important to use prudently this antimicrobial class, trying to avoid their unjustified off-label use.
Lines 336-338: Do the Authors realize that the difference between a dosage of 5 and 25 mg/kg is very large? Why was not a dosage within this range also tested?
Lines 342-348: The Authors stated that “In Germany enrofloxacin is currently approved for the treatment of digestive and respiratory diseases in companion birds and poultry caused by a variety of bacterial species at a dosage of 10 mg/kg body weight”.
Just some very important questions:
- Is it the specific veterinary medicinal product employed in this study approved in Germany for the use in companion birds and poultry? With what therapeutic indications? By what route of administration?
- If Enrofloxacin is currently approved in Germany at the dosage of 10 mg/kg BW, why did the Authors choose to administer 25 mg/kg BW?
- Even if Enrofloxacin concentrations in aqueous humor were lower than in serum, what would be the scientific basis for recommending improper use of such an important antimicrobial, given that the study was done in healthy animals?
- Additionally, given that the MICs used to discuss the results of this study are mainly referable to the epidemiology of another continent (which has a completely different policy for the use of antimicrobials) and very different from each other, do the Authors think it prudent to encourage the use of this dosing regimen?
Lines 371-374: The Author stated “These data indicate that enrofloxacin penetrates to a higher extent into the aqueous humour in the chicken compared to mammals due to anatomical and functional differences in the blood-retinal barrier”.
Please, remove this sentence since in this study it was not assessed the ability of enrofloxacin to penetrate the blood-retinal barrier of birds at the dosages administered to mammals.
Author Response
In this study, the Authors determined the pharmacokinetic profile in serum and aqueous humour of enrofloxacin following a single intramuscular administration in chicken at the dosage of 25 mg/kg BW. Specifically, the Authors investigated the feasibility of this regimen to achieve clinically effective concentrations for the possible treatment of intraocular infections with sensitive bacteria.
The manuscript is well-written, with a proper experimental study design to provide pharmacokinetic information but not proper to provide therapeutic indications.
The novelty is clear, but the importance of this investigation due to the implications within the topic of antimicrobial stewardship needs to be further discussed.
To be this manuscript considered for publication, the Reviewer needs some major concerns to be sincerely addressed by the Authors.
Simple Summary
Lines 23-24: Please, remove this sentence or change it since the study was not performed on animals affected by eye infections, so is not possible to state that enrofloxacin is useful for the treatment of these conditions.
We agree with the statement and removed the last sentence of the simple summary.
Introduction
Lines 59-61 and 67-68: “Because of its favourable characteristics, enrofloxacin, a fluoroquinolone antimicrobial agent, is frequently used in avian medicine”.
Do the Authors think using enrofloxacin for many types of systemic bacterial infection can be considered a prudent antimicrobial use? Enrofloxacin is a and veterinary critically important antimicrobial, and its use should be limited to infections sustained by bacteria resistant to first- and second-choice active compounds.
The addressed topic of critically important antibiotics is clearly very important. For this reason, we have decided to add these issues in the revised version of the manuscript (Lines 108-111).
The present study aims to contribute to a responsible and targeted use of enrofloxacin in the avian patients. Enrofloxacin is reserved for veterinary use only.
Significantly fewer antibiotics are available to the veterinarian for treatment of birds compared to mammals for various reasons (incompatibilities, lack of regulatory approval, lack of pharmacokinetic studies, unfavorable pharmacokinetic properties, such as a very short half-life etc.). The described anatomical peculiarities of the eye, especially concerning the blood-ocular barrier, further limit the applicable antibiotics, since most antibiotics are not able to cross the blood-ocular barrier. Enrofloxacin is one of the few antibiotics expected to be effective on the eye. Our study aims to verify whether enrofloxacin as an antibiotic of critical importance in avian could be suitable at all for the treatment of ocular infections in birds.
Other options would have been chloramphenicol or marbofloxacin. However, chloramphenicol is considered obsolete due to various properties, such as a health risk to the user. Marbofloxacin is also a 2nd generation fluoroquinolone and thus has the same limitations as enrofloxacin.
Materials and Methods
Line 127: How did the Authors choose the dosage? Why was it not decided to have two treatment groups, one with a registered dosage and one off-label?
This is indeed an interesting question. As mentioned already at the response to reviewer 2, the study was conducted on chickens as a pharmacological model. However, the results of the study are expected to benefit primarily ornamental birds and birds of prey. In Germany enrofloxacin is currently approved for poultry at an oral dose of 10 mg/kg body weight. For ornamental birds, it is approved for the treatment of infections of the gastrointestinal and respiratory tract at a single daily i.m. dose of 20 mg/kg body weight (Baytril® 25mg/ml Injection solution). Because Baytril® 25mg/ml Injection solution is not labeled for treatment of intraocular infections, we decided to use a dose for our study, that takes into account the limiting function of the blood-aqueous barrier on drug diffusion. Indeed, our results verified that enrofloxacin concentrations in aqueous humour are significantly lower than those in serum. In the current literature, dosages of up to 30 mg/kg BW twice daily are recommended for ornamental birds, depending on the infectious agent and site of infection (Hawkins et al: Birds. In Exotic Animal Formulary, 5th ed.; Carpenter, J.W., Marion, C.J., Eds.; Elsevier Health Sciences: 2018; pp. 178-179., Doneley, B. Formulary. In Avian Medicine and Surgery in Practice: Companion and Aviary Birds, Second ed; Doneley, B., Eds.; CRC Press, Taylor and Francis Group: New York, 2016: pp. 425-454). The chosen dosage of 25 mg/kg once daily is thus within these dosage recommendations available in the scientific literature and in a range where no side effects are expected.
This issue is now stated more clearly in the revised version of the manuscript (Lines 348-354).
Line 137-141: Were the Authors not concerned that alterations in the cardiovascular pattern related to general anesthesia, particularly systemic pressure, might affect the outcome of drug delivery?
It is right that a general anesthesia could affect the outcome of the drug delivery. General anesthesia generally causes a drop in heart rate and blood pressure. From this, it could be anticipated that in the non-anesthetized patient even higher drug concentrations could be found in serum and aqueous humor and that the results in our study are rather underestimated by the general anesthesia. However, animals were only anesthetized for 5min in prior to blood sampling and collection of aqueous humour. Only for time points 2 and 4 min, drug delivery was during anesthesia. This was necessary for animal welfare reasons, because paracentesis in avian patients is not possible only with local anesthesia of the cornea compared to humans or mammals. We added these aspects to the discussions section on lines 300-306.
Lines 153-155: Out of curiosity: since the animals were to be euthanized, did the Authors also consider conducting a residual depletion study?
As already mentioned at Reviewer 2, drug depletion studies are usually done with twice the therapeutic dose. From this, evaluation of the fate of enrofloxacin would have required an additional set of experiments, which was not a topic of this study. There is indeed some literature on this issue, although not at a dosage of 25mg/kg (de Assis et al, 2016, Slana et al, 2014, Slana et al, 2017, Morales et al, 2015, San Martin et al, 2007), In addition, although this study was conducted on chickens as a pharmacological model, the results of the study are expected to benefit primarily ornamental birds and birds of prey, where the residue issue is less relevant.
Lines 198-199: Why did the Authors choose two different extraction methods for the two matrices (i.e., serum and aqueous humour)?
There were two issues that had to be considered for analysis of the aqueous humor samples. First, we had only 50 µL instead of 100 µL of sample material per workup. In addition, the concentrations of the analytes, especially ciprofloxacin, were relatively low in aqueous humor and even lower than in serum. With the conventional method, the ciprofloxacin would have been under the limit of detection. With the immunoaffinity columns, more concentrated solutions could be prepared (dilution by a factor of 10 instead of 40, i.e. 4-fold more concentrated). In addition, the volumes of aqueous humour were too small (0,05 ml) to be analyzed directly by LC-MS/MS, so purification with immunoaffinity columns had to be performed as an intermediate step.
Discussion
Lines 326-328: If these antimicrobial “are hardly able to pass the intact barrier”, why did the Authors choose an experimental model with only healthy animals? Instead of a critically important antimicrobial within an experimental animal model it was possible to try other active compounds in sick animals with an altered blood-retinal barrier.
It is right that inflammatory changes caused by infection can change drug penetration. Therefore, we have added another section to refer to this aspect in the discussion (Lines 382-389).
The aim of our study was to collect data to answer the question whether enrofloxacin could be considered as a systemic therapeutic agent in avian ophthalmic patients. Initial studies on pharmacokinetic parameters are generally conducted in clinically healthy animals. Many bacterial infections induce inflammation and consequently pathologic ocular changes in birds. From this, the use of clinically healthy animals most probably represents the most stringent condition for the penetration of enrofloxacin in the eye, since the birds had an intact blood-ocular barrier. It has been assumed that disease might lead to a breakdown of the blood-ocular barrier. It can thus be speculated that even in diseased animals with breakdown of the blood-ocular barrier enrofloxacin will penetrate well into the aqueous humour.
We agree that the effectiveness of enrofloxacin in a clinical setting of eye diseases is an important issue and has to be investigated in future population kinetics studies.
Lines 329-330: “Fluoroquinolones are drugs of choice in systemic treatment of intraocular infections in man”. Exactly, and in particular for this reason it is important to use prudently this antimicrobial class, trying to avoid their unjustified off-label use.
We agree with this statement. In avian medicine, there are no antibiotics approved for systemic treatment of bacterial intraocular infections to date. Therefore, the aim of our study was to collect data for a prudent and reasonable use of enrofloxacin and to help avoiding unjustified use. We clarified this aspect in lines 106-111.
Lines 336-338: Do the Authors realize that the difference between a dosage of 5 and 25 mg/kg is very large? Why was not a dosage within this range also tested?
The present study is a baseline study in which we limited ourselves to one dosage. In Germany enrofloxacin is approved with an oral dose of 10 mg/kg BW for poultry and an intramuscular dose of 20mg/kg BW for pet birds for treatment of infections of the gastrointestinal and respiratory tract. In case of more complicated infections, the manufacturer itself suggests the use of a higher dose. Because of the reasons mentioned above, we consider intraocular infections to be complicated. Dosage recommendations available in the literature range up to 30 mg/kg twice daily for pet birds. Our chosen dosage of 25 mg/kg body weight is within the recommended range and in a range, where no side effects are yet expected. We added further information regarding this aspect in lines 353-355.
Lines 342-348: The Authors stated that “In Germany enrofloxacin is currently approved for the treatment of digestive and respiratory diseases in companion birds and poultry caused by a variety of bacterial species at a dosage of 10 mg/kg body weight”.
Just some very important questions:
Is it the specific veterinary medicinal product employed in this study approved in Germany for the use in companion birds and poultry? With what therapeutic indications? By what route of administration?
An injectable preparation which is approved in food-producing animals was selected for the trial. There is no enrofloxacin product in Germany, which is approved for both, pet birds and poultry. Likewise, no injectable preparation is available for poultry. Furthermore, a product with an active ingredient concentration of 50 mg/ml was chosen in order to keep the injection volume low and thus avoid extensive muscle damage to the animals in consequence of the application. The chickens in this experiment served as a pharmacokinetic model for other avian species and were therefore untethered from regulatory restrictions on the use of certain pharmaceuticals. However, the experiments were conducted in accordance with the German animal welfare regulations and under permission of the German authorities (reference number ROB 55.2-2532.Vet_02-19-165). Ocular diseases are not yet included in the therapeutic indications of enrofloxacin, so its use for such an indication is off-label. Our study aimed to collect data for a justified and prudent use of enrofloxacin in avian ocular diseases. There are no antibiotics in Germany approved so far for the treatment of bacterial eye diseases in birds. We commented on this in lines 353-359.
If Enrofloxacin is currently approved in Germany at the dosage of 10 mg/kg BW, why did the Authors choose to administer 25 mg/kg BW?
As already mentioned above, we have noted that the information on the approval of enrofloxacin in ornamental birds and poultry was not clearly stated in our paper. This issue is now stated more clearly in the revised version of the manuscript (Lines 353-359). The dosage of 25 mg/kg body weight was chosen because of the presence of an additional biological barrier, the blood-ocular barrier. This expectation was finally confirmed by the results of our study.
Even if Enrofloxacin concentrations in aqueous humor were lower than in serum, what would be the scientific basis for recommending improper use of such an important antimicrobial, given that the study was done in healthy animals?
We understand the concerns that exist with the use of a critically important antibiotic such as enrofloxacin, but this study aims to reduce improper use of antibiotics and to provide data for rational and scientifically based use of enrofloxacin in avian medicine. Indeed, there are several reasons why we used enrofloxacin for this study: First, fluoroquinolones, besides chloramphenicol, are the most lipophilic antibiotics known to cross the blood-ocular barrier. Second, the antibiotic spectrum of enrofloxacin covers most of the bacteria commonly causing ocular infection. Finally, enrofloxacin is available for avian medicine and is currently the antibiotic of choice for treatment of intraocular infections.
The effect of intraocular infections on the blood-ocular barrier is not exactly known. It was assumed to lead to a breakdown the blood-ocular barrier. It has, however, to be expected that the effect, among other things, essentially depends on the causal pathogens. Therefore, it was important to find a dose that can still be effective when the blood-ocular barrier is still intact despite infection. We therefore selected a higher dosage, which we considered to be promising for the penetration of the blood-ocular barrier, but which is still below the maximum dosage recommended in the literature, at which, for example, clinical side effects could occur.
Additionally, given that the MICs used to discuss the results of this study are mainly referable to the epidemiology of another continent (which has a completely different policy for the use of antimicrobials) and very different from each other, do the Authors think it prudent to encourage the use of this dosing regimen?
This is of course an important issue, and we are sorry for not being able to present more appropriate data at this time. Currently there is no better data available. Furthermore, we do not regard the availability of data for another continent as a disadvantage, because we did not want to verify the usability of enrofloxacin in ocular diseases of birds in Germany only but provide a generally valid approach.
Lines 371-374: The Author stated “These data indicate that enrofloxacin penetrates to a higher extent into the aqueous humour in the chicken compared to mammals due to anatomical and functional differences in the blood-retinal barrier”.
Please, remove this sentence since in this study it was not assessed the ability of enrofloxacin to penetrate the blood-retinal barrier of birds at the dosages administered to mammals.
We agree that the chosen wording is not accurate and propose to adjust the sentence as it is now written in the manuscript (Lines 382-386).
We would like to thank the reviewer for the careful review and for the helpful comments to further improve the manuscript.
Round 2
Reviewer 3 Report
The reviewer thanked the authors for their responses.
In this form, the manuscript is now substantially improved and the risk of misinterpretation by inexperienced readers has been significantly lowered.